# Influence of the Microbiome Metagenomics and Epigenomics on Gastric Cancer

**DOI:** 10.3390/ijms232213750

**Published:** 2022-11-09

**Authors:** Precious Mathebela, Botle Precious Damane, Thanyani Victor Mulaudzi, Zilungile Lynette Mkhize-Khwitshana, Guy Roger Gaudji, Zodwa Dlamini

**Affiliations:** 1Department of Surgery, Steve Biko Academic Hospital, University of Pretoria, Hatfield 0028, South Africa; 2School of Medicine, University of Kwa-Zulu Natal, Durban, KwaZulu-Natal 4013, South Africa; 3SAMRC Research Capacity Development Division, South African Medical Research Council, Tygerberg, Cape Town 7501, South Africa; 4Department of Urology, Level 7, Bridge C, Steve Biko Academic Hospital, Faculty of Health Sciences, University of Pretoria, Private Bag X323, Arcadia 0007, South Africa; 5SAMRC Precision Oncology Research Unit (PORU), DSI/NRF SARChI Chair in Precision Oncology and Cancer Prevention (POCP), Pan African Cancer Research Institute (PACRI), University of Pretoria, Hatfield 0028, South Africa

**Keywords:** gastric cancer (GC), metabolites, microbiome, *H. pylori*, dysbiosis, epigenomics, personalized therapy, obesity, asbestos-induced GC, inflammasome

## Abstract

Gastric cancer (GC) is one of the major causes of cancer deaths worldwide. The disease is seldomly detected early and this limits treatment options. Because of its heterogeneous and complex nature, the disease remains poorly understood. The literature supports the contribution of the gut microbiome in the carcinogenesis and chemoresistance of GC. Drug resistance is the major challenge in GC therapy, occurring as a result of rewired metabolism. Metabolic rewiring stems from recurring genetic and epigenetic factors affecting cell development. The gut microbiome consists of pathogens such as *H. pylori,* which can foster both epigenetic alterations and mutagenesis on the host genome. Most of the bacteria implicated in GC development are Gram-negative, which makes it challenging to eradicate the disease. Gram-negative bacterium co-infections with viruses such as EBV are known as risk factors for GC. In this review, we discuss the role of microbiome-induced GC carcinogenesis. The disease risk factors associated with the presence of microorganisms and microbial dysbiosis are also discussed. In doing so, we aim to emphasize the critical role of the microbiome on cancer pathological phenotypes, and how microbiomics could serve as a potential breakthrough in determining effective GC therapeutic targets. Additionally, consideration of microbial dysbiosis in the GC classification system might aid in diagnosis and treatment decision-making, taking the specific pathogen/s involved into account.

## 1. Introduction

Stomach cancer, also known as gastric cancer (GC) is one of the key causes of cancer-related deaths worldwide, with slowly changing 5-year survival rates (5–69%) [1,2]. High mortality rates due to the disease being difficult to detect in its earliest stages result in most cases being detected at a nonresectable stage. The incidence of the disease is generally higher in males than in females [3]. The urgency for specific biomarkers for early detection of the disease is crucial, as this will offer a better prognosis. The identified biomarkers could be useful for the development of targeted therapy. Currently, the treatment offered to cancer patients includes chemotherapy, radiotherapy, surgery, immunotherapy and hormonal therapy [4]. Nonetheless, these treatment options have their limitations, while the tumor microenvironment (TME) and other cancer factors foster resistance to therapy.

Cancer therapy is recommended and implemented depending on the severity and type of cancer. Systematic chemotherapy is the mainstay of treatment for GC patients with an advanced stage of the disease [5]. Despite the efforts put in the advancement of chemotherapy, drug resistance by GC cells remains a challenge due to the disease’s complexity and multifactorial mechanisms which require a thorough understanding of the different “omics” and how they interact with each other. The microbiome casts a long shadow over GC and influences most if not all the “omics” of cancer. The role of the microbiome has been reported to contribute to the hallmarks of cancer including inflammation, cellular invasion and metastasis, apoptosis resistance, and metabolic reprogramming [6,7,8,9,10]. Existing data shows that microbiota undoubtedly regulates tumor initiation, cancer progression, and resistance to therapy through chronic inflammation induced by pathogenic agents, dysbiosis, and metabolism rewiring. The emergence of metabolomics brought an enhanced understanding of the relationship between metabolic regulation in cancer, and how this is influenced by bacteria-derived metabolites, genetic and epigenetic factors, and how they are manipulated by the microbiome in GC and other cancers [11,12,13,14].

Microbes produce metabolites that have a modulating effect on the host epigenome leading to carcinogenesis [15]. According to Engstrand and Graham, 2020, data on the GC microbiome is not compelling enough to show its role in cancer pathogenesis beyond what is ascribed to *H-pylori* infection dynamics [16]. The authors hereby review the role of the microbiome GC carcinogenesis by identifying the association between GC microbiomics risk factors, host metagenomics, metabolomics, and epigenomics. The review would also attempt to ascertain the microbiome as one of the root challenges of GC carcinogenesis and progression, and could therefore be considered as part of the key therapeutic targets of the disease.

## 2. Gastric Cancer Subtypes

Distinguishing the subtypes of any cancer is fundamental in aiding in the identification of biomarkers, exploring treatment options and determining prognosis. Lauren’s criteria and the 2010 World Health Organization (WHO) classification system are the commonly used methods to classify GC. Lauren’s criteria categorize cancer into two main subtypes, the intestinal and diffuse types of adenocarcinomas, and it is mostly used as it provides information on the clinical management and outcome of the disease [17]. The WHO classification system histologically classifies the types into mucinous, papillary, tubular, poorly cohesive carcinomas, and other rare or mixed histological types such as Paneth cell and mixed adeno-neuroendocrine carcinomas [18]. However, regardless of the use of these two systems, recurrence of the disease still occurs in more than 70% of the patients, with more than 80% of them dying within two years following curative surgery [19].

Gastric cancer was initially classified using pathohistological and anatomical characteristics, however, these were ineffective in guiding therapy with minimal improvement in disease outcome over time. Clinical and molecular features seem to be more promising in guiding the choice of effective treatment. Molecular subtypes are identified mainly by using genomics. In 2011, Tan et al. introduced genomic intestinal (G-INT) and genomic diffuse (G-DIF) subtypes with distinct histology, gene expression patterns, biological pathways, and prognosis [20]. These subtypes are somewhat associated with Lauren’s classification. This is owed to the inherent clinically heterogeneous nature of the disease on account of the varying molecular characteristics of cancer cells [21]. Thanks to the next generation sequencing (NGS) technology, the molecular principles and large amounts of cancer genetics data can be explored, enabling us to identify novel targets for precision medicine as cancer subtypes get unfolded. Table 1 shows the different types of GC molecular classifications, associated genes, and the prognosis of each type. As more data is obtained, the classification of GC will be more precise, and personalized therapy will be better guided. This review provides evidence of the contribution of the microbiome in GC carcinogenesis and progression. The inclusion of the microbiome as part of the therapeutic intervention more in particular by taking into consideration the type of pathogen implicated in carcinogenesis is suggested, thus aiding in biomarker and therapy designs.

## 3. The Link between Gut Microbiome and Gastric Cancer Risk Factors

One of the proposed cancer prevention strategies is risk factor (RF) reduction. Treatment of the underlying RF can therefore reduce the risk of developing cancer or aid in the treatment of cancer resulting from RF predisposition. A systemic review by Yusefi et al. reported a total of 52 GC RFs which were identified and classified according to 9 categories influenced by familial genetics, lifestyle, environment, medication, and exposure to toxins [32]. These categories can be further grouped into two sub-categories; genetic and modifiable, with genetic factors being hereditary while modifiable ones are acquired through lifestyle and can be changed. The most common RFs for GC include *Helicobacter pylori (H. pylori)* infection, metabolic syndrome, an increased salt intake with a diet low in fiber, as well as male gender (two-fold increase in males than females) [1,4,33,34,35]. Although GC is more common in males, some subtypes such as the MSI and CIMP-H tumors are more prevalent in females [23,28,29]. It was initially thought that GC affects people of an older age (50 to 70 years), however recent findings show an increased incidence in younger individuals [36,37]. The modifiable RFs often lead to epigenetic alterations, and examples of these include toxins, diet, obesity, infection, etc. [38]. Figure 1 shows how various RFs can lead to GC.

### 3.1. Obesity

The International Agency for Research on Cancer (IARC) regards obesity as the second leading cause of cancer after smoking [39]. About 3–9% of all cancers are approximated to stem from obesity, and GI cancer with obesity origin has the worst prognosis [40,41]. One of the risk factors for obesity includes a high fat and sodium diet, which alters the gut microbiota composition, resulting in gut microbial dysbiosis [42]. Dysbiosis regulates the susceptibility and initiation of many gut malignancies [43]. Kim et al. found that *Fusobacterium* was enriched in fecal samples of metabolically unhealthy overweight and obese individuals [44]. This shows that bacteria are a common RF in obesity and GC.

### 3.2. Diabetes

Diabetes is considered an important contributing factor in GC development, and it is postulated that this is due to shared RFs. These include obesity, a higher infection/reinfection rate, and a lower eradication rate of *H. pylori*, as well as the chronic use of medication [45]. Additionally, increased salt intake may cooperate with *H. pylori* infection in the induction of GC and progression. However, a 2022 metanalysis showed no association between diabetes and GC risk in the grading of *H. pylori* infection and other shared RFs [46]. The authors concluded that diabetes may be associated with excess cardia GC risk.

### 3.3. Acid Reflux-Related Disorders

Several studies have indicated the association between gastroesophageal reflux disease (GERD) and GC [47,48,49,50]. The overall 5-year survival rate of gastric cardia adenocarcinoma is reported at approximately 31% [51]. Two subtypes of gastric cardia cancer exist; one with GERD origin and the other associated with atrophic gastritis [52]. Misumi et al. defined gastric cardia carcinoma as “a lesion with its center located within 1 cm proximal and 2 cm distal to the esophagogastric mucosal junction” [53]. In a study by Ye et al, it was reported that the risk of developing gastric cardia adenocarcinoma persisted following anti-reflux surgery [50]. This shows that GERD can lead to long-term effects on the stomach mucosa. Some of the risk factors of GERD are obesity and a diet low in fibre, which can have an effect on the gut microbiome [54]. Generally, acid reflux is linked to gut microbiome dysbiosis [55]. A retrospective study by Polat and Polat reported that 82.5% of 1437 GERD patients had *H. pylori* infection with 1–3 severity score [56], bearing in mind that *H. pylori* infection is a common RF for GC and GERD.

### 3.4. Chronic Infection and Inflammation

Infection with pathogenic microbiota leads to the upregulation of inflammatory markers such as cytokines and other secretory proteins. Cytokines such as tumor necrosis factor (TNF), interleukin- 1 (IL-1) and IL-6 expressed within the TME induce cell invasion, metastasis, angiogenesis, growth, and anti-apoptotic effects [57,58,59]. Colonization of *H. pylori* in the stomach leads to chronic inflammation via the activation of Wnt/β-catenin and other pathways that get activated by the bacteria’s virulence, which further permits the bacteria to survive and thrive in the gut [60,61]. The Wnt/β-catenin signaling pathway is crucial in modulating key cellular processes contributing to carcinogenesis, such as apoptosis, metastasis, proliferation, and genetic stability [62]. Moreover, Wnt/β-catenin has been implicated in pancreatic cancer chemoresistance [63].

The *H. pylori* commonly infects the stomach, leading to chronic diseases such as peptic ulcer, gastritis, and gastrointestinal (GI) cancers such as GC. The stomach’s naturally acidic environment assists in preventing infection by pathogens. The *H. pylori* bacteria can maneuver this acidic environment and alter the overall profile of the gastric microbiome [64,65]. There are three mechanisms that the bacteria utilize to alter the GI microbial profile to favor their survival [66]. This includes the employment of enzymes such as ureases which help the bacteria to buffer the acidic pH of the stomach [67]. Secondly, the *H. pylori* infection effects changes on the cell cycle of gastric epithelial cells, resulting in the elevated expression of p21 and p53 proteins and leading to gene mutations [68,69]. In addition, the infection can lead to abnormal molecular signaling pathways [27]. According to Rossi et al., genomics and proteomics cannot be used to monitor response to therapy [70]. However, a study by Goodman et al. provided evidence that cell-free DNA (cfDNA) can be used to monitor response to chimeric antigen receptor T-cell (CAR-T) therapy in patients with a certain type of B-cell lymphoma [71]. Similarly, the Lewis protein CA-19 is routinely used as a gold standard marker for monitoring response to pancreatic cancer therapies [72,73]. The potential of these “omics” in the area of therapeutics is limited and not well understood.

Gram-negative bacteria including *H. pylori* are highly resistant to numerous drugs and antibiotics due to the protection provided by their outer membrane [74]. The chronic inflammatory response induced by *H. pylori* predisposes the mucosal cells to carcinogenesis. In a prospective, double-blind, placebo-controlled, randomized trial published in 2018 by Choi et al., it was observed that GC patients who had either endoscopic resection of early GC or high-grade adenoma, after receiving *H. pylori* ablation therapy, had lower metachronous GC rates compared to their counterparts who received a placebo [75]. Later on, the same team conducted a randomized trial that was published in 2020 where they evaluated the treatment of *H. pylori* in first-degree relatives of GC patients [76]. Their results showed that the treatment lowered the risk of developing GC by 55% when compared to the placebo group. Moreover, the risk of developing GC was lowered in 73% of participants who were confirmed for *H. pylori* ablation than those who had persistent infection of the bacteria. These results confirm the potential use of RFs as therapeutic targets for cancer therapy.

The *H. pylori* bacteria initiates GC by causing the DNA to replicate faster due to the chronic inflammation incited by the organism, and this leads to mutagenesis and genomic instability. Inflammation is a hallmark of cancer which plays a key role in all three carcinogenesis stages [77]. The virulence factors of *H. pylori,* such as the cytotoxin-associated gene A (cagA), are responsible for its chronic inflammation properties. CagA functions as an oncoprotein and can trigger MAPK signaling of host cells, leading to persistent inflammation and uncontrollable proliferation [27]. Additionally, the MAPK pathway is responsible for chemoresistance in pancreatic cancer and GC cells [78,79]. It is postulated that cagA travels through the type IV secretion systems (T4SS) upon contact with the host cell and this triggers the endocytosis of the protein (Figure 2) [80]. The protein can activate the MAPK/ERK pathway in two ways: by direct binding in a phosphorylation-independent state or through recruiting the phosphatase SHP2 [81,82]. The SHP2 protein plays a crucial role in the pathologic activity of cagA and can independently modify ERK signals autonomous of Ras [8].

The MAPK/ERK is also known as the RAS/RAF/MEK/ERK signaling pathway and plays a major role in regulating cell differentiation, proliferation and survival. This pathway is interlinked with the PI3K/Akt/mTOR pathway and can cause compensatory signal transduction in cases where the other is compromised [83]. The coupled inhibition of the two pathways has been effective in tumor stasis and overcoming drug resistance of GI tumor cells [84,85]. An elevated expression of the Ras protein is positively associated with increased Akt protein levels. Thus, PI3K/Akt/mTOR is an alternative pathway to Ras/Raf/MEK/ERK for EGFR signaling [86]. This may affect the efficacy of anticancer treatment, and therefore this must be considered when developing novel anticancer therapies. The bacteria can also initiate GC through aberrant DNA methylation, which will be discussed in more detail later in the review. Moreover, the expression of DNA mismatch repair (MMR) genes MutS and MutL are decreased in *H- pylori*-positive gastric mucosa [87].

The antibiotic metronidazole functions by interacting with the DNA of the target organisms (Gram-negative bacteria) breaking down DNA strands and causing the loss of DNA integrity and the ultimate inhibition of protein synthesis [88]. Metronidazole gets activated upon reduction by the protein ferredoxin (Figure 3). The concentration gradient created upon reduction increases the diffusion of metronidazole into the bacterial cell and cytotoxic free radical generation [88]. The drug has been shown to be successful in treating *H. pylori*; however, the bacteria has evolved to be resistant to it and can only be effective when in combination with esomeprazole and amoxicillin [89,90].

Another type of Gram-negative bacteria, Fusobacteria, is considered a RF for GC. The *Fusobacterium* spp., predominantly *F. nucleatum,* are frequently found in abundance in GC, pancreatic and colorectal tumors compared to non-cancerous tissues [43,91,92]. *F. nucleatum*-positivity has been linked to overall worse survival in Lauren’s diffuse type of GC and MSI-high status of colon cancer [91,93]. It is therefore safe to assume that *F. nucleatum* predisposes individuals to the MSI-high subtype of GC. Just like *H. pylori*, Fusobacteria can be eradicated with metronidazole therapy. The bacteria is known to be highly sensitive to the drug [94]. Apart from bacteria, other viruses such as Epstein-Barr virus (EBV), which is sometimes referred to as human herpesvirus 4 (HHV4), raises the risk of GC by a factor of 18 times and the EBV-associated GC (EBVaGC) is observed more in males than in females [95]. EBVaGC contributes to approximately 10% of all GC cases worldwide and is more common in the early stages of the disease [96]. Other viruses with potential association with GC include the human papillomavirus (HPV), hepatitis B virus (HBV), John Cummingham virus (human polyomavirus 2) and human cytomegalovirus [97]. More research on these viruses is required to determine their role in GC pathogenesis.

## 4. Other Microbes Implicated in GC Pathogenesis

Carcinogenesis describes the process of cancer formation which stems from irreversible genetic alterations or interruptions due to internal and external factors. It is a multistage molecular process involving (i) initiation, (ii) promotion and (iii) progression [98,99]. The microbes can either directly affect the cells and lead to carcinogenesis or tamper with the body’s cellular pathways to support its growth and sustainability. The gut microbiome, which is also known as the human second genome, plays a major role in the pathogenesis of GI cancers including colorectal, pancreatic, liver and gastric [43,100,101,102]. Bacterial and viral pathogens negatively influence the host’s genomic stability and integrity by the destruction of DNA strands, thereby initiating tumor development [103]. Approximately 95% of the human body’s microbiota resides in the gut and the microbes generally assist in maintaining the balance between health and disease [104]. There appears to be microbiome dysbiosis in most cancers, and this has been found to aggravate tumorigenesis. Although the microbiome is implicated in a number of cancers, the exact mechanisms by which they lead to cancer is still controversial. This is due to the low biomass of the microbiota in the TME, making it challenging to study them further [105]. Thanks to omics studies, this challenge can be overcome, as they shed light on the role of the gut microbiome in cancer pathology, prevention, and therapy [106].

### 4.1. The Boas-Oppler Bacillus

The lactic acid bacillus (lactobacillus), which is commonly called the Boas-Oppler Bacillus, dates back to 1895 when Izmar Isidor Boas and Bruno Oppler described the role of these Gram-positive bacteria in GC [107]. In their study, the researchers discovered that the bacillus was present in abundance in the gastric juices of 95% of GC individuals included in the study. This has been observed to be common, especially in patients with an advanced stage of the disease [108]. Lertpiriyapong et al. reported that infection of insulin–gastrin (INS-GAS) transgenic mice with *L. murinus* ASF361 led to the development of gastric neoplasia via the upregulation of oncogenes and pro-inflammatory genes [109]. Lactobacilli produce lactic acid/lactate which plays a huge role in the Warburg effect, a hallmark of cancer. Additionally, the Lactobacilli play a role in the production of excessive amounts of N-nitroso compounds (NOCs), which are carcinogenic and predispose *H. pylori*-free individuals to GC [110,111]

### 4.2. Mycoplasma

The study of the role of mycoplasma infection in cancer development dates way back to the 1950s [112]. They are Gram-negative bacteria that belong to the class Mollicutes [113]. The bacteria are commonly known for causing infections of the ear, respiratory system, lungs, urogenital tract and also to cause sexually transmitted infections (STIs) [114]. The well-studied pathogenic species include *Ureaplasma urealyticum, M. fermentans, M. penetrans, M. hominis, M. genitalium, M. pneumoniae*, *M. hyorhinis,* etc. The *M. hyorhinis* species is implicated in the development of GC [7,112,115,116,117]. Although mycoplasma have been detected in GC biopsies, the infection is not considered a RF for the disease [118]. Research has shown that mycoplasma cause inflammation, which instigates cancer initiation and progression [119,120].

A p37 lipoprotein located on the outer membrane of *M. hyorhinis* has been proven to play a key role in tumorigenesis [116,121,122]. The p37 protein heightens the expression of inflammation-associated genes such as vascular cell adhesion molecule 1 *(Vcam1)*, *IL-6*, *IL-1*, and lipocalin 2 *(LCN2)* [119]. Additionally, p37 promotes cell invasiveness by blocking contact inhibition, and this has been observed in melanoma, gastric and prostate carcinomas [116,121,122]. Gong et al. demonstrated that p37 promotes the metastasis of human GC and lung cancer cells through the activation of matrix metalloproteinase-2 (MMP-2) and EGFR/PI3K/AKT/ERK pathways [7]. Another mechanism by which the mycoplasma promotes metastasis is via the accumulation of β-catenin and the activation of its Wnt signaling pathway [121,123]. Moreover, metastasis in GC by *M. hyorhinis* can be initiated through activation of the NACHT, LRR, and PYD domains-containing protein 3 (NLRP3) [10]. The NLRP3 is an inflammasome critical in caspase-1 modulated inflammation in response to pathogenic organisms [124]. Because these organisms lack peptidoglycan and are Gram-negative, this makes them extremely resistant to antibiotics [114]. The *M. hyorhinis* infection has been linked with the diffuse-type GC with a higher infection rate in advanced stages (TNM III/IV) than in earlier stages of the disease [117]. On the other hand, *M. hyorhinis* can cause chronic infections which induce chromosomal instability, and one can classify this under the TCGA CIN subtype of GC [10,23]. The age group of GC patients who are more likely to be infected with mycoplasma is the elderly [117].

## 5. Compounds Linked with GC Induction

### 5.1. Contribution of Microbes in Asbestos-Induced GC

Amosite, actinolite, chrysotile, anthophyllite, crocidolite, and tremolite are the six types of asbestos of which chrysotile (white asbestos) is the most abundant (99%) and is also exceedingly hazardous and lethal [125,126]. However, this does not mean that the other types are less harmful, as they also possess toxicity to some extent [127]. The link between asbestos and GI cancers was first demonstrated by Selikoff et al. in 1960, then in 2012, a review by Kim et al. reported that among all GI cancers, GC is the one that is greatly linked with asbestos exposure [128,129]. Oksa et al. summarized the association between GC and asbestos exposure and concluded that the risk of developing GC is directly proportional to asbestos exposure with the risk ranging from 15% to 20% [130]. In a study by Patel-Mandlik and Millette, an olive baboon that was fed chrysotile was discovered to have asbestos fibers deposited in its stomach while other pieces were able to relocate to most neighboring tissues except for the small intestine [131]. This shows that the fibers cannot be digested and can remain in the stomach for longer periods of time before their excretion [132]. Because of its strength and chemical properties, the material does not get digested or broken down, and the exposure elicits scarring and irritation, resulting in inflammation of the tissue [132]. Data shows that prolonged asbestos exposure leads to chronic inflammation and cellular stress, which activates the MAPK pathway and related transcription factors leading to immune response gene expression [133]. The gut microbiome plays a vital role in modulating immune homeostasis and GC inflammation. However, its association with asbestos in cancer is poorly reported.

Stanik et al. evaluated the ability of *L. casei* and *L. plantarum* to biologically break down white asbestos fibers [134]. The bacteria were successful due to their ability to produce lactic acid, which contains hydrogen ions that can remove magnesium ions from the crystalline structure of the asbestos fibers. A study by Seshan showed that when chrysotile is exposed to strong acids like those of the stomach or water, the physical and chemical properties of the asbestos change as the magnesium is lost from the asbestos [135]. There is evidence that shows that *L. plantarum* is capable of preventing *H. pylori*-induced inflammation of the gastric mucosa and restores balance to the gut microbiome, which is altered during such an infection [136]. Pretreatment with these bacteria was able to slow down the expression of inflammatory cytokines and cell infiltration. Similarly, *L. casei* has an anti-cancer effect, as it is able to inhibit the mTOR and NF-kB signaling pathways, thereby leading to the cellular apoptosis of GC cells [137]. Figure 4 shows how *L. casei* and *L. plantarum* could potentially prevent GC carcinogenesis. From this information, we can deduce that these two Lactobacilli species can be used to aid in the treatment of GC, more in particular one with an asbestos origin.

Asbestos fibers can be ingested and pass through the esophagus and lodge into the stomach lining. These fibers do not pass through to the small intestines where they could possibly go through the process of excretion but remain in the stomach long enough to induce GC.
(A)*L. Plantarum* has the ability to block *H. Pylori*-induced inflammation that is associated with GC development.(B)*L. Casei* bacterium downregulates pro-oncogenic signaling pathways (NF-kB and mTOR) thus inhibiting cancer development and progression.(C)This pair of bacteria can alter the chemical and structural properties of white asbestos by the removal of magnesium ions, a process that could be explored as preventative therapy in individuals exposed to asbestos fibers or as therapeutic intervention in asbestos-induced GC.

### 5.2. Enterobacteriaceae and Nitrosamines Production

Nitrosamines are carcinogenic N-nitroso compounds which can nest in the stomach and are produced when amines react with nitrites. They can either be ingested as an outside source or produced from ingested food with the help of certain bacteria [138]. Foods that contain nitrites include processed meats, fish, fried bacon, beverages, and cheese [138]. Cigarettes and E-cigars also release some nitrosamines called tobacco-specific nitrosamines (TSNAs) when inhaled and can result in DNA damage and mutagenesis [139]. Exposure has been correlated with GC RFs such as diabetes and pathogenesis to the mammary glands, leading to breast cancer [140,141,142]. In breast cancer TSNAs actively bind nicotinic acetylcholine receptors (nAChRs), activating its signaling pathways. The alpha7 receptor (α7nAChR) is an oncoprotein that plays a role in both the initiation and progression stages of breast cancer carcinogenesis [143]. The estrogen receptor-positive type of breast cancer carcinoma has been shown to express the α7nAChR in high levels [143]. Nitrosamines have been reported in lung cancer as activators of the NF-kB and PI3K/AKT signaling pathways, which are pivotal in cell proliferation [144].

There is a positive correlation between ingestion from nitrosamine sources and GC [145,146,147]. The nitrosamine hypothesis dates back to the 1950s and paved the way for research investigating the role of the gut microbiome and GC until the focus shifted towards *H. pylori’s* role in chronic inflammation. Nitrate reductases are secreted by Gram-negative bacteria called *Enterobacteriaceae.* These enzymes catalyze the conversion of nitrate to nitrite [148]. These bacteria play a key role in nitrosamine production in the gut. In a study by Sarhadi et al., *Enterobacteriaceae* was found to be abundant in fecal samples of different GC types [149]. Similar findings were reported by Liu et al., who detected *Escherichia* and *Streptococcaceae* in abundance in GC patients [150]. Qin et al. reported an abundance of *Enterobacteriaceae* in diabetic patients, one of the risk factors of GC [151].

## 6. The Role of the Gut Microbiome on the Epigenomics of GC

Since the human genome coexists with the gut microbiome it is only fitting that a form of crosstalk exists between the two genomes in order to maintain homeostasis [152]. The metabolites released by certain gut microbiota have regulatory effects that induce epigenetic modifications, thereby influencing gene expression. Although it is not yet clearly understood how these metabolites modify host gene expression, metabolites produced by gut microbiota including biotin, short-chain fatty acids (SCFAs), amino acids, etc., have been implicated in these alterations. It is worth noting that metabolic reprogramming and epigenetic remodeling are closely linked hallmarks of cancer and mutually regulate each other [153]. Epigenetic alterations appear phenotypically and do not originate from changes in the DNA sequence [154]. These alterations can be trans-generationally inherited, and genes involved in oncogenic pathways are often affected by epigenetic modifications rather than mutations [155]. The alterations begin to occur during the preliminary stages of tumorigenesis and are potential therapeutic targets for GC as they are more specific [156].

Although epigenetic changes can be reversed, prolonged exposure to epigenetic altering agents that lead to the retention of a gene regulatory protein often result in permanent changes [157]. In the case of *H. pylori*, a form of class I carcinogen can potentially induce carcinogenesis through epigenetic modifications in the form of hypermethylation-silencing of numerous tumor suppressor genes (TSGs) [158]. TSGs’ aberrant DNA methylation as a result of *H. pylori* infection takes place on the promoter CpG island, and this represses the transcription of corresponding downstream genes which may result in irreversible TSG inactivation [159]. Similarly, EBV-positive GC has the CpG island methylator phenotype [160]. A study on gerbils showed that GI cancers could possibly be prevented by suppressing DNA methylation induction [161]. Infection with the HBV increases the risk of developing GC, especially in patients without familial history of the disease [102]. Infection with this virus affects the methylation of a number of TSGs including *p16* and lead to uncontrollable cellular proliferation [13]. Chronic infection with HBV can also lead to irreversible methylation. There has not been much reported on the effect of HBV treatment in GC patients, and more studies are needed to further explore this area of research. Figure 5 indicates how epigenetic-involved infection with the above-mentioned pathogens can result in tumorigenesis.

## 7. The Effect of Proton Pump Inhibitors on the Gut Microbiome

Accumulating evidence shows that prolonged use of proton pump inhibitors (PPIs) increase the risk of developing GC [162,163,164]. The PPIs are generally used to treat acid reflux related disorders, GERD, peptic ulcers, Zollinger-Ellison syndrome (ZES), pancreatitis, and esophagitis [165,166,167]. Most of the above-mentioned diseases are RFs of GC, thus supporting the potential link between prolonged use of these drugs and their contribution to GC carcinogenesis. The mode of action for PPIs is to block gastric hydrogen potassium ATPase in an acidic environment [168]. In the process, these inhibitors cause gut microbiome dysbiosis via gastric-acid suppression, indulging the profusion of pathogenic bacteria [55,163].

## 8. Gut Microbiome in Metabolic Rewiring

Metabolomic data shows that there is a link between the metabolic and epigenetic mechanisms and that these mechanisms are differentially regulated in normal and malignant cells due to their intrinsic metabolic variances [169]. The gut microbiome is a key regulator of metabolism, which means dysbiosis can disrupt homeostasis and lead to metabolic rewiring. Metabolic rewiring forms part of the hallmarks of cancer and is crucial for tumors as they require increased amounts of energy to supply cells that are constantly growing and proliferating at an increased rate. Reprogramming is made possible through dysbiosis, the stimulation of oncogenes, and mutated metabolic enzymes [170]. The key pathways involved in this rewiring include mitochondrial biogenesis, glutaminolysis, anaerobic glycolysis, lipid metabolism, and other biosynthetic pathways, most of which are regulated by specific bacteria [171].

Normal cells generally utilize oxidative phosphorylation (OXPHOS) to generate ATP for the energy required for cellular function, whereas cancer cells depend on anaerobic glycolysis, an observation termed the Warburg effect. Just as in other tumors GC exhibits the Warburg effect, a phenomenon whereby tumor cells display an increased uptake of glucose and its conversion to lactate through glycolysis in the cytoplasm instead of traditional OXPHOS in the mitochondria [172]. OXPHOS and glycolysis are interlinked and cooperate to maintain energetic homeostasis; however, for a long period of time, it was believed that carcinogenesis leads to the permanent impairment of the mitochondrial OXPHOS, automatically switching cancer cells to the glycolytic pathway [172,173]. This metabolic alteration affects the availability of a number of metabolites including acetyl-CoA. The tumorigenic effect of glycolysis is that it enables tumor cells to limit the availability of glucose on the TME and this impairs immune cells’ function against the tumor [174]. An enzyme called pyruvate kinase M2 (PKM2) which catalyzes phosphoenolpyruvate (PEP) to pyruvate in the rate-limiting step of glycolysis is very crucial in cell metabolism and the Warburg effect. The dimeric PKM2 is a trait that all proliferating cells have in common and favors lactate production in tumor cells, and is highly oncogenic [175].

Short-chain fatty acids are metabolites of bacteria that play a vital role in gene regulation, and the most common ones are acetate and butyrate. The Warburg effect can be reversed by a phenomenon called the “Butyrate Paradox”, where cancer cells switch from glycolysis to OXPHOS upon butyrate exposure. Certain types of healthy gut microbiota produce the beneficial compound butyrate through anaerobic fermentation of high-fiber foods. The bacteria known to produce butyrate include Eubacterium, Clostridium, Ruminococcus and Coprococcus [176]. There is a lack of data on studies comparing the quantity of these bacteria in GC patients and their healthy counterparts. This information might be useful in determining if GC might result as a consequence of their deficiency.

A study by Bouwens et al. showed that a high-fiber diet effectively lowered the risk of colon cancer [177]. Mounting evidence indicates that the activity of metabolic enzymes including PKM2 in colon cancer are altered through the direct binding of butyrate, consequently reversing the Warburg effect and enhancing chemotherapy [169,178]. Butyrate binding inhibits the phosphorylation of the PKM2 enzyme and encourages its tetramerization (Figure 6). In cancer cells, butyrate accumulates as a result of the Warburg effect, as these cells depend on glucose as their main source of energy. Accumulated butyrate functions as a histone deacetylase (HDAC) inhibitor, which terminates cell cycle progression through altered gene expression, and this enhances the response to chemotherapy [179].

## 9. Gut Microbiome and its Products in Gastric Cancer Therapy

### 9.1. Bacteriotherapy

Bacteriotherapy is a promising field of cancer therapy that utilizes genetically modified bacteria, or a live but weakened form, as well as bacteria-derived substances or particles such as peptides that have anticancer properties [180]. Bacteriocins are secondary metabolites in the form of peptides released by bacteria such as the Lactobacilli and have antibacterial properties which inhibit the growth of other bacteria [181]. These metabolites can be divided into four categories; class I (mw: <5 kDa) which are also known as lantibiotics, class II (mw: <10 kDa) which are thermostable, class III (mw: >30 kDa) which are heat-labile and able to disrupt cell membranes, and lastly class IV, which consists of proteins with lipid or carbohydrate components [182]. Bacteriocins are effective in inhibiting the growth of antibiotic-resistant strains as well as pathogenic bacteria, thus maintaining gut homeostasis [183,184]. This is not the only anti-cancer trait that these peptides have. They can also induce cytotoxicity and apoptosis, making them attractive for cancer therapy. Ou et al. hypothesized that an increase in colorectal cancer risk is due to the disproportion of the health-promoting and carcinogenic metabolites [185]. This might be true for all GI cancers, as they are all influenced by metabolism.

### 9.2. Butyrate-Based Therapy

Sodium butyrate is capable of inducing GC cell apoptosis through the elevation of death-associated protein kinase (DAPK1/2) and caspase 3 expression, and depression of Bcl-2 [186]. Panebianco et al. showed that butyrate supplementation heightens kidney and liver damage markers, as well as inducing apoptosis and inhibiting cell growth of pancreatic cancer cells both in vitro and in vivo [187]. As previously mentioned, butyrate can actively bind the PKM2 enzyme and reverse the Warburg effect (Figure 6). In a study by Geng et al., treatment of colonocytes with butyrate was able to increase the efficacy of chemotherapy and repair DNA synthesis [178]. Another molecule that has the same effect on PKM2 as butyrate is the heat shock protein 40 (HSP40) chaperone. This novel discovery was made by Huang et al., where HSP40-PKM2 binding resulted in the downregulation of the PKM2 protein levels and in turn regulated glucose metabolism by inhibiting glycolysis and cancer cell development [188].

Wang et al. investigated the role of PKM2 in GC, and it was discovered that there is an overexpression of the protein in GC patients and that this overexpression was linked to poor prognosis and clinicopathologic parameters of the disease [189]. In the same study, the in vitro and in vivo knockdown of *PKM2* displayed inhibition of tumor progression in GC cell lines and xenograft mice, respectively. Additionally, their results showed that when *PKM2* is treated with short hairpin RNA (shRNA) which gets processed to small interfering RNA (siRNA), one of the pillars of epigenetic modifications halts tumor growth and progression, along with cell migration and proliferation. Various studies showed that *PMK2* expression’s impediment by shRNA elevates the sensitivity of tumor cells to treatment with docetaxel, and that cells with silenced *PKM2* were more prone to undergo apoptosis [88,190,191].

### 9.3. F. Nucleatum as Potential Therapeutic Target

Postoperative adjuvant chemotherapy is often administered to lower the risk of cancer recurrence. *F. nucleatum* infection persists post-neoadjuvant chemoradiotherapy (nCRT) in colorectal cancer, and this is linked to high relapse rates, which is a major cause of tumor recurrence [9]. Yu et al. demonstrated that *F. nucleatum* promotes chemoresistance by regulation of autophagy through loss of miR-18a* and miR-4802, and this loss is influenced by the activation of the TLR4/MYD8 [192]. The TLR4/MyD88 pathway leads to the downstream activation of the NF-κB pathway, which plays a substantial role in cancer development and progression [193,194]. In addition, the authors suggested that the quantity of *F. nucleatum* be determined for individual patients so that the regimens could be personalized and given in combination with chemotherapy [192]. The *F. nucleatum* bacterium therefore plays an important role in cancer prognosis. An ongoing interventional study in Shanghai, China has enrolled 294 colorectal cancer patients in postoperative stages II/III [195]. They aim to use oral metronidazole to reduce the abundance of *F. nucleatum* in patients having a high bacterial count of the organism to explore whether the drug can improve the potency of postoperative chemotherapy in patients with colorectal carcinoma. An additional study on metronidazole was a clinical proof-of-concept interventional study in Zealand, Denmark [196]. This study was based on the observation that *F. nucleatum*-positive xenograft mice displayed decreased tumor load and intratumoral profusion of the bacteria following oral administration of metronidazole [197]. Their study explores the effect of combining fosfomycin with metronidazole for the treatment of colon biofilms and adenomas. A few studies have shown that biofilm formation and *F. nucleatum* are mostly linked to right-sided colon cancers and adenomas [198,199].

### 9.4. Anti-Mycoplasma Therapy

Inhibitors of MMPs, ERK have been proven to block the p37-induced invasion in GC cells, and this can be manipulated for potential therapy in *M. hyorhinis* patients [7]. Either treatment of GC cells with the β-catenin inhibitor XAV939 or its knockdown was able to halt metastasis and therefore these could further be explored for the treatment of GCs [121].

### 9.5. CRISPR/Cas9

Genomic therapeutic clinical trials that are ongoing include clustered regularly interspaced short palindromic repeats (CRISPR), which have the ability to move us from hype to reality by providing insight into essential therapeutic gene targets, mechanisms of tumorigenesis, and to allow provocative studies in drug resistance [200,201]. CRISPR makes up the genomic editing system and the hallmark of the native bacterial defense mechanism which has been adapted in cancer immunotherapy [202,203,204]. Accidentally discovered by Ishino and his colleagues in the 1980s, the CRISPR is made of two genetic units, CRISPR loci which contain spacers and repeats, and operons of *cas* genes [202,205]. The CRISPR/Cas9 system is categorized into two classes (I and II) and the well-known CRISPR/Cas9 forms part of the type II of class II which has a relatively simpler structure and can easily be studied [206]. The key components of the CRISPR/Cas9 system are the guide RNA (gRNA) and the Cas9 protein. Though the gRNA of prokaryotes can only recognize viral DNA, its synthetic form is generated with the ability to target any gene sequence for editing [207]. A recent paper stated that The Sichuan University’s West China Hospital in Chengdu was the first to enroll patients for a non-randomized clinical trial study on CRISPR cancer therapy from 2016 to 2018. In this study they had two primary end points, safety and feasibility, as well as efficacy as the secondary endpoint [208].

Lu et al. transfused gene-edited T-cells to non-small cell lung cancer (NSCLC) patients by using the CRISPR/Cas9 technique. The transfused cells had the protein called programmed cell death protein 1 (PD-1) edited such that it does not bind the PD-L1 ligand of cancer cells and inherently prevent immunosuppression. Their results showed that the CRISPR/Cas9 gene-edited T-cells are safe with minor side effects and are feasible for use in clinical settings. Similar findings were observed by Su et al., where EBVaGC xenograft mice were transfected with PD-1-disrupted T-cells using the CRISPR/Cas9 system and displayed an increased immune response and cancer cell death [209]. Thus, this type of immune checkpoint targeted therapy can be trialed on GC patients, particularly the EBV and MSI subtypes, since they are associated with *PD-L1* overexpression [28]. It is advised that future clinical trials should utilize more advanced gene editing approaches to improve therapeutic efficacy [208]. An example is the retron library recombineering (RLR) approach. Unlike CRISPR/Cas9, RLR functions without cutting DNA and can be applied to huge populations of cells within a short space of time [210]. So far, the system has been performed in bacterial cells and displays more than 90% efficiency. Though this technology is still at its infancy stages, its efficacy as far as gene editing therapy is concerned looks promising.

### 9.6. PI3k/Akt/mTOR Signaling Pathway Targets

The phosphatidylinositol 3-kinase (PI3K)/Akt/mammalian target of the rapamycin (mTOR) pathway is a key therapeutic target of cancer cells, as its components are observed to be activated in most cancers including GC. It is worth noting that a few PI3K inhibitors have been endorsed by the US Food and Drug Administration (USFDA) for the treatment of advanced metastatic breast cancer as well as chronic lymphocytic leukemia and small lymphocytic lymphomas [211]. This pathway plays a huge role in regulating cell proliferation, tumor cell growth, angiogenesis, migration, survival and therapeutic resistance [212]. Genetic mutations detected in GC correlate with altered signals involving the PI3K/Akt/mTOR pathway, and overactivation has been observed in up to 40% of tumor types [213]. An example of these genes is the PI3K’s p110 catalytic subunit *PIK3CA*, which is the third most frequently mutated gene in GC following tumor protein 53 *(TP53)* and AT-rich interactive domain-containing protein 1A (*ARID1A)* and has been implicated as an oncogene in various cancers [23,214,215]. Mutations of this gene activate the PI3K/AKT/mTOR signaling pathway and other downstream signaling pathways, which leads to tumorigenesis [216].

The cause of mutations of genes involved in the PI3K/AKT/mTOR pathway is not clear; however, it is debatable that they might occur as a consequence of dysbiosis. This is based on data that shows that probiotic bacteria of the family Lactobacillaceae, which are known to restore eubiosis, an interspecies balance of the microbiome, can prevent cancer through modulation of the involved pathways and the immune response [217]. *L. casei* and *L. fermentum* form part of the natural microbiome of the oral, GI, and vaginal tracts in humans [218]. They have been reported to prevent GC by reducing the expression levels of NF-κB and IκB, which decreases the phosphorylation of PI3K and Akt, thereby inhibiting the growth of GC cells [137,219]. Probiotics containing these bacteria could be administered to GC mice and the response to therapy monitored.

### 9.7. Microbial Ablation

The higher the microbial diversity in the gut microbiome, the more favorable the outcomes of cancer treatment will be [220]. As mentioned earlier, *H. pylori* infection is a top RF for GC. Following *H. pylori* ablation treatment, the gut microbiota profiles were altered and a decline in ghrelin levels were observed in *H-pylori*-positive patients [221]. In a study by Aykat et al., KC mouse models’ inflammation by oncogenic *Kras* led to fungal dysbiosis characterized by *M. globosa*, promoting pancreatic tumor progression through the activation of the mannose-binding lectin (MBL)-C3 cascade while mycobiome ablation shielded the mice against oncogenic progression [222]. The ablation of pathogenic organisms involved in GC is one of the systems that could be explored further in pre-clinical studies with the intention of translating these studies into clinical trials.

## 10. Conclusions and Future Considerations

Evidently, GC carcinogenesis is induced by infection with different microbial pathogens resulting in the emergence of dysbiosis. Dysbiosis restoration therapy could pave the way for improved management and alleviation of the disease. Gram-negative bacteria are at the forefront of cancer initiation, development, and resistance to therapy. The fact that these pathogens have a protective layer makes them challenging to treat. One of the most effective treatments for such bacteria, metronidazole, can be used in combination with other antibiotics to deliver a more desirable effect. The GC categories can also be classified according to the type of carcinogenic organism detected in an individual. This can be considered when designing a personalized cancer treatment. Combinatorial therapies can therefore be designed with a specific antibiotic depending on an individual’s disease profile.

The TME microbiome influences certain pathways in the cancer hallmark and studying them might provide insight into the mechanisms of GC development and progression. Most of these pathways are interconnected and undergo compensatory signal transduction, necessitating the development of therapeutic drugs with combinatory elements or targets. Risk factor management and treatment are crucial in the management of GC. Some of the RFs are common in both RF diseases, and cancer and can be managed to decrease the chances of developing GC and other GIT cancers. Most GI cancer cells respond in a similar manner, and therefore clinical trials performed in one type of cancer could be reproduced in another. The toxicity of the effective existing treatments should be lowered by altering treatment regimens or by adding cytoprotective agents such as misoprostol and sucralfate, which may protect the body from some of the side effects during clinical trials. The challenge of the complexity of GC can be overcome through the understanding of the different molecular subtypes of the disease and appreciating that this could be achieved through the integration of multi-omics.

## Figures and Tables

**Figure 1 ijms-23-13750-f001:**
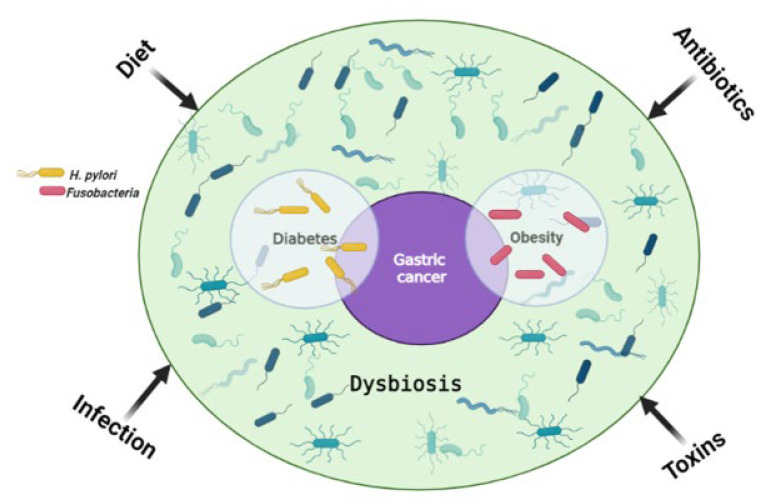
The main risk factors for gastric cancer. Environmental factors influence the gut microbiome and can lead to dysbiosis, one of the main causes of tumorigenesis. *H. pylori* infection is a shared risk factor between gastric cancer and diabetes, with diabetes being a risk factor for gastric cancer on its own. Similarly, *Fusobacteria* are a common risk factor for obesity and gastric cancer, with obesity on its own being a risk factor for gastric cancer. Created with BioRender.com. (accessed on 20 September 2022).

**Figure 2 ijms-23-13750-f002:**
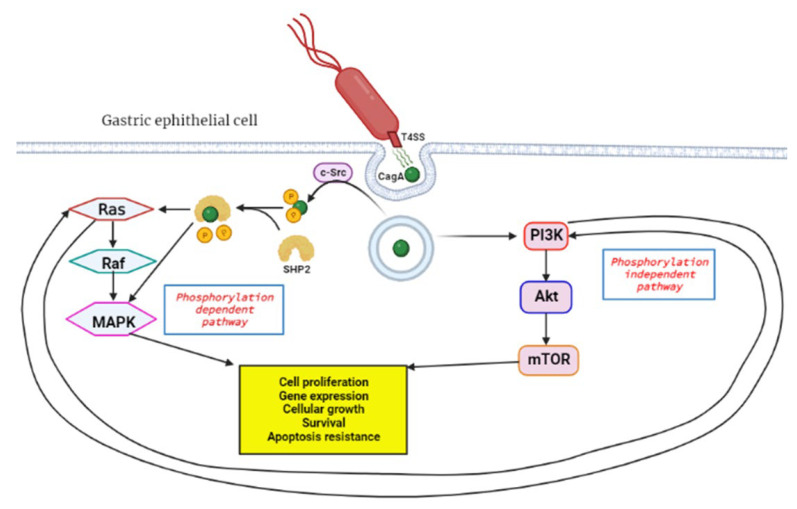
Activation of RAS/RAF/MEK/ERK pathway by *H. pylori* cagA oncoprotein. Upon contact with the gastric epithelial cell membrane, the bacteria’s T4SS system releases cagA through a channel, and this triggers endocytosis, a process where proteins get engulfed into the cell. In the phosphorylation dependent pathway c-Src, tyrosine kinase phosphorylates cagA, followed by SHP2 phosphatase cleavage of the phosphate groups from cagA. This leads to downstream activation of the RAS/RAF/MEK/ERK signal transduction pathway which favors tumorigenesis. The phosphorylation independent pathway is the PI3K/Akt/mTOR, which gets activated by cagA and results in products that induce tumorigenesis. Created with BioRender.com. (accessed on 27 September 2022).

**Figure 3 ijms-23-13750-f003:**
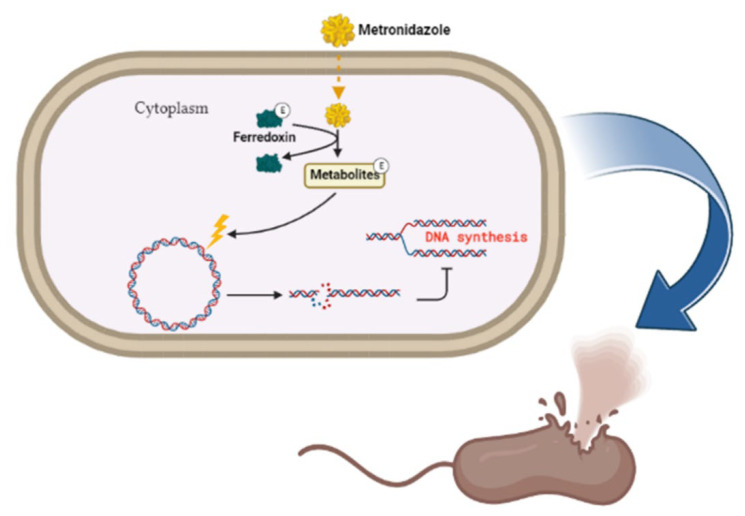
Metronidazole’s mode of action. The inert drug enters susceptible bacterial cells through passive diffusion. Metronidazole is activated through its reduction by ferredoxin. Upon activation of the drug, a concentration gradient is formed, and this favors the increased uptake of the drug into the organism, thus elevating its antimicrobial effect. DNA damage subsequently leads to protein synthesis inhibition and consequent apoptosis. Created with BioRender.com. (accessed on 27 September 2022).

**Figure 4 ijms-23-13750-f004:**
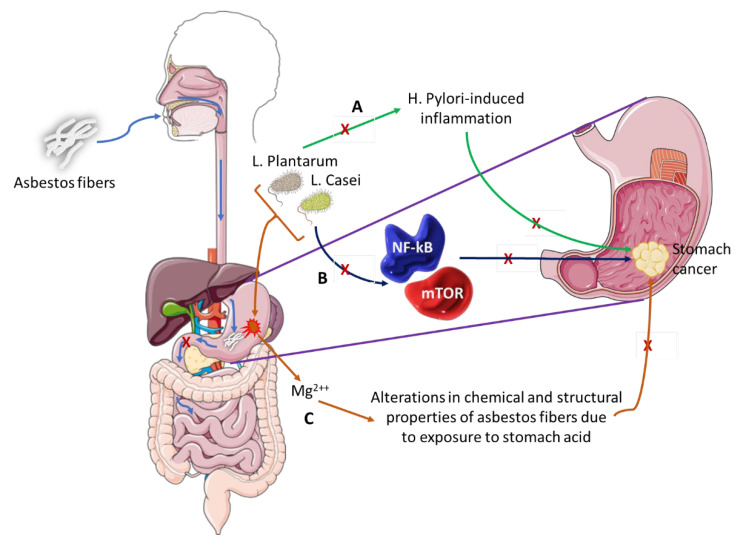
The role of microbiota in asbestos-induced GC.

**Figure 5 ijms-23-13750-f005:**
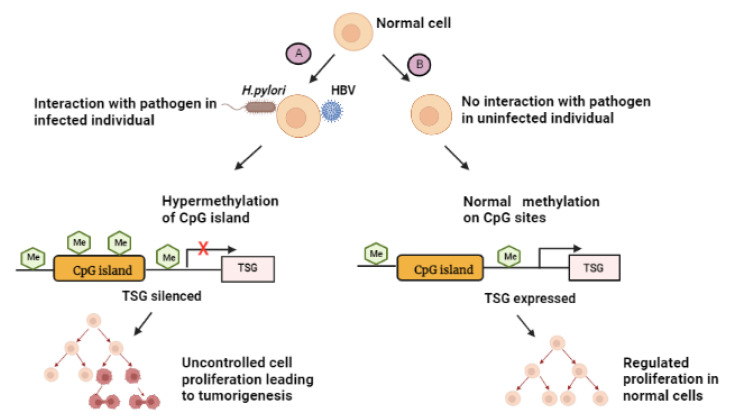
Summary of the effect of CpG island methylation as a result of pathogen infection. (**A**): Pathogens come into contact with the infected host cell, leading to events that cause hypermethylation of the CpG island to occur. Methylation of CpG islands of the promoter results in TSG silencing and uncontrolled cell proliferation, thus inducing gastric cancer. “**X**” represents the inhibition of TSG activation. (**B**): In the absence of pathogens, normal methylation takes place. Cell division is under surveillance as normal transcription continues and TSGs are expressed. Me - methylation, TSG - tumor suppressor gene. Created with BioRender.com. (accessed on 27 September 2022).

**Figure 6 ijms-23-13750-f006:**
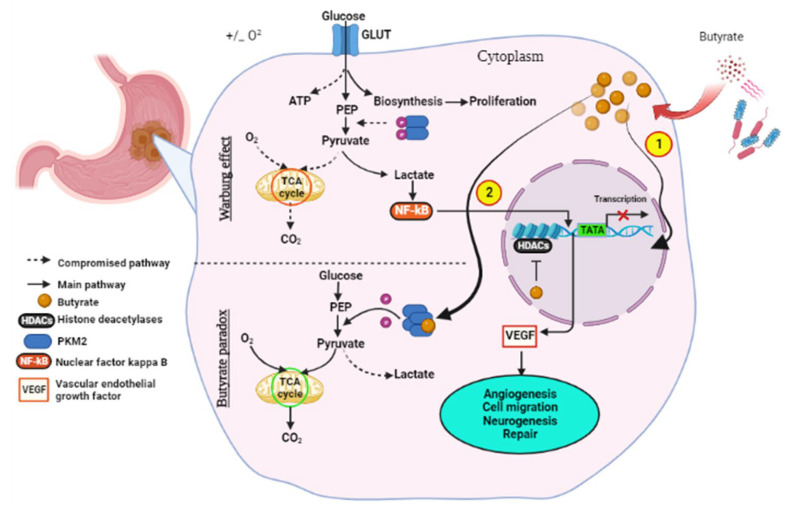
The anti-cancer effect of butyrate. Butyrate molecules are produced through the fermentation of fiber by bacteria. The molecules accumulate in cancerous gastric epithelia as the Warburg effect addicts these cells to the glucose metabolism and the increased production of lactate. Lactate triggers events that lead to VEGF upregulation, subsequently leading to cancer progression. Butyrate helps fight off cancer through the Butyrate Paradox phenomenon in two ways: (**1**) by travelling to the nucleus where it functions as a HDAC inhibitor which terminates cell cycle progression through altered gene expression; and (**2**) by reversing metabolism from anaerobic glycolysis to conventional OXPHOS through binding PKM2, altering it to a more active dephosphorylated tetrameric form. This favors energy production through the Krebs cycle. VEGF—vascular endothelial growth factor, HDAC—histone deacetylase. Created with BioRender.com. (accessed 27 September 2022).

**Table 1 ijms-23-13750-t001:** Different types of GC molecular classifications.

Classification	Subtypes	Prognosis	Associated Genes	Ref.
Intrinsic subtypes	G- INT	Better overall survival	*FUT, LGALS4, CDH17*	[20]
G-DIF	Poor	*AURKB, ELOVL5*
Lei subtypes	Proliferative	Short disease-free survival	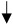 *TP53*	[22]
Metabolic	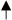 *TP53*
Mesenchymal	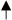 *TP53*
TCGA	EBV- positive	Best	*PIK3CA, JAK2, PD-L1/2, BCOR*	[23]
MSI	Moderate with no adjuvantchemotherapy response	*PIK3CA, ERBB2/3, EGFR, PD-L1, MLH1, TP53*
GS	Worse	*CDH1, RHOA*
CIN	Moderate	*SMAD4, APC, TP53*
ACGA	MSI- high	Best with lowest recurrence frequency	*ARID1A, MTOR,* *KRAS, PIK3CA, ALK, PTEN*	[24]
MSS/EMT	Worse with highest recurrence frequency	*CDH1*
MSS/TP53+	Moderate	*APC, ARID1A, KRAS, PIK3CA, SMAD4*
MSS/TP53-	Moderate	*ERBB2, EGFR, CCNE1, CCND1, MDM2, ROBO2, GATA6, MYC*
Combined TCGA and ACRG	EBV- positive	Best	*PIK3CA, JAK2, PD-L1/2, BCOR*	[25,26]
MSI- high	Best with lowest recurrence frequency	*ARID1A, MTOR,* *KRAS, PIK3CA, ALK, PTEN*
GC with aberrant E-cadherin	*	*
GC with aberrant p53 expression	*	*
GC with normal p53 expression	*	*
CIMP	CIMP-H	*	EBV-associated	[27,28,29,30,31]
CIMP-L	*	*
CIMP-N	Worse survival	*

Abbreviations: genomic instability (G-INT), genomic diffuse (G-DIF), chromosomal instability (CIN), genomic stable (GS), microsatellite instability (MSI) and Epstein- Barr virus (EBV), microsatellite stable/epithelial-mesenchymal transition (MSS/EMT), CpG island methylator phenotype-High/Low/Negative (CIMP-H/L/N), not clearly established (*), increased (
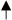
) and decreased (
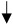
) mutations.

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
