# Peer review of "Influence of the Microbiome Metagenomics and Epigenomics on Gastric Cancer"

_ijms, 2022, doi:10.3390/ijms232213750_

Round 1
Reviewer 1 Report
A valuable and intriguing paper, but lacking of consideration of al least one crucial and non-negligible topic: no consideration of the relationship between gastric microbioma and nitrosamines.
About this item, e.g. see:
- Engstrand L, Graham DY, 2020; Song P, Wu L, Guan W, 2015;
- under an historical perspective, Schmahl D, 1978; Issemberg P, 1976.
In my humble opinion, useful some attention to the relationship between asbestos adn gastric cancer (e.g. see Fang YJ et al Increased risk of gastric cancer in asbestos-exposed workers. Int J Environ Res Pb Health 2021).
Author Response
Dear Reviewer,
Thank you for the time spent reviewing the manuscript.
Kindly see the attached response letter.

Reviewer 2 Report
In the present study, authors want to explore the role of microbiome in Gastric Cancer, and this is an interesting topic, however, it seems that authors fail to reach this purpose. Below is my main commands:
The language is good, and the paper reads smoothly. But the title authors used is “Influence of the Microbiome Metagenomics and Epigenomics on Gastric Cancer”, so authors should mainly discuss the role of microbiome in the occurrence, development, treatment and prognosis of Gastric Cancer, and how to take use of this in the treatments of Gastric Cancer. But in the whole paper, authors mostly listed the known knowledge on the GC and its mechanisms, which well written, but lack novelty and and is boring.
So, this paper need a major revision to deeply explore the role of microbiome (stomach or gut) on the GC, and its potential application in clinic trails.
Author Response
Dear Reviewer
Thank you for the time spent reading the manuscript.
Kindly see the attached response letter.

Round 2
Reviewer 1 Report
The paper has been adequately improved; it doesn't introduces deeply novel approaches to the topic, but provides a clear and useful overall look to the reader.
Reviewer 2 Report
Acceptable